# HSF-1 activates the ubiquitin proteasome system to promote non-apoptotic developmental cell death in *C. elegans*

Maxime J Kinet[†], Jennifer A Malin[†], Mary C Abraham, Elyse S Blum, Melanie R Silverman, Yun Lu, Shai Shaham*

Laboratory of Developmental Genetics, The Rockefeller University, New York, United States

**Abstract** Apoptosis is a prominent metazoan cell death form. Yet, mutations in apoptosis regulators cause only minor defects in vertebrate development, suggesting that another developmental cell death mechanism exists. While some non-apoptotic programs have been molecularly characterized, none appear to control developmental cell culling. Linker-cell-type death (LCD) is a morphologically conserved non-apoptotic cell death process operating in *Caenorhabditis elegans* and vertebrate development, and is therefore a compelling candidate process complementing apoptosis. However, the details of LCD execution are not known. Here we delineate a molecular-genetic pathway governing LCD in *C. elegans*. Redundant activities of antagonistic Wnt signals, a temporal control pathway, and mitogen-activated protein kinase kinase signaling control heat shock factor 1 (HSF-1), a conserved stress-activated transcription factor. Rather than protecting cells, HSF-1 promotes their demise by activating components of the ubiquitin proteasome system, including the E2 ligase LET-70/UBE2D2 functioning with E3 components CUL-3, RBX-1, BTBD-2, and SIAH-1. Our studies uncover design similarities between LCD and developmental apoptosis, and provide testable predictions for analyzing LCD in vertebrates.

*For correspondence: shaham@rockefeller.edu

[†]These authors contributed equally to this work

**Competing interests:** The authors declare that no competing interests exist.

## Introduction

Animal development and homeostasis are carefully tuned to balance cell proliferation and death. Cell death not only counters cell production, but also supports morphogenesis and tissue sculpting, and destroys cells that could be harmful, such as autoreactive cells in the immune system or genetically abnormal cells that may promote tumorigenesis. Developmental and homeostatic cell elimination are not passive processes but rather follow a highly coordinated, genetically encoded program (*Green, 2011*). A major goal has been to identify the molecular basis of the programs controlling regulated cell demise in development. One such program, apoptosis, has been studied extensively over the past four decades. Some proteins that promote apoptosis, such as BCL2 and FAS, are mutated in human disease (*Fisher et al., 1995*; *Rieux-Laucat et al., 1995*; *Tsujimoto et al., 1985*), indicating that apoptosis contributes to normal human physiology.

Nonetheless, caspase-dependent apoptosis does not account for many cell death events that take place during normal animal development. For example, in the moth *Manduca sexta*, salivary gland and body muscle remodeling is apparently caspase-independent and the ultrastructural morphology acquired by dying cells is non-apoptotic (*Haas et al., 1995*). Similarly, mice homozygous for knockout alleles of key apoptotic genes, including caspase-3, caspase-9, Apaf-1, or Bax and Bak, can survive to adulthood (*Honarpour et al., 2000*; *Kuida et al., 1998*; *Lindsten et al., 2000*), a surprising observation given the prevalence of cell death in murine development. Indeed, nearly half of

**eLife digest** Embryos make numerous new cells as they develop, but also destroy many cells to remove the faulty ones and to ensure that tissues grow to the right size and shape. This deliberate form of cell death must be precisely regulated to prevent too many cells or healthy cells, from being destroyed. Understanding the molecular mechanisms that govern cell death is therefore important for understanding normal development and also human disease.

One well-studied process that leads to cell death is called apoptosis. This process carefully dismantles and breaks down the components of a cell, but does not seem to account for all cell death that occurs during animal development. Recently another developmental cell-death pathway, called the linker-cell-type death, was discovered in a small roundworm called *Caenorhabditis elegans*. This pathway appears to work in mammalian cells as well, and may help to break down nerve fibers that are not needed. However, many of this pathway's component parts remained unknown.

Kinet, Malin et al. have now used a combination of genetics and cell biology in *C. elegans* to uncover the components of linker-cell-type death and to investigate how they interact. The results of these studies revealed a hierarchy of genetic interactions that governs this pathway in *C. elegans*. One protein called HSF-1 plays a particularly important role. This protein is a transcription factor and it binds to, and regulates, the activities of various genes. HSF-1 usually works in cells to protect them from stress, but Kinet, Malin et al. showed that it instead promotes linker-cell-type death by activating a molecular machine, called the proteasome, that breaks down proteins. The experiments also revealed two proteins (called BTBD-2 and SIAH-1) that may be important for shuttling specific proteins for degradation by the proteasome.

Three signalling pathways that regulate important developmental processes also regulate the activation of linker-cell-type death. Kinet, Malin et al. propose that these signalling pathways do so by working together to activate HSF-1, which in turn activates the genes that lead to the destruction of cells by the proteasome.

A future challenge is to understand in more detail how the more recently discovered cell death pathway actually kills cells. Further work could also explore how HSF-1, a protein that normally protects cells, is transformed into a cell-killing protein.

spinal cord motor neurons generated during vertebrate development are normally deleted, and this process occurs unabated in the absence of caspase-3 or caspase-9 (*Oppenheim et al., 2001*). While caspase-independent non-apoptotic processes may play key roles in developmental cell death, little is known about their molecular underpinnings. To date, none of the non-apoptotic cell death pathways that have been described have a role in normal development (*Zhou and Yuan, 2014*).

The *Caenorhabditis elegans* linker cell provides direct evidence that caspase-independent non-apoptotic cell death pathways operate during animal development. This male-specific gonadal leader cell guides the elongation of the gonad and *vas deferens* during development, and then dies near the cloaca, presumably to facilitate fusion of the *vas deferens* with the cloacal sperm-exit channel (*Kimble and Hirsh, 1979*). Linker cell death still occurs in the absence of the main apoptotic caspase, CED-3, and even in animals lacking all four *C. elegans* caspase-related genes (*Abraham et al., 2007*; *Denning et al., 2013*). Other canonical apoptosis genes are also not required, nor are genes implicated in autophagy or necrosis (*Abraham et al., 2007*). Consistent with these genetic observations, the morphology of a dying linker cell, characterized by lack of chromatin condensation, a crenellated nucleus, and swelling of cytoplasmic organelles, differs from the morphology of apoptotic cells (*Abraham et al., 2007*). Intriguingly, cell death with similar features (linker cell-type death [LCD]; *Blum et al., 2012*) has been documented in a number of developmental settings in vertebrates (*Pilar and Landmesser, 1976*) and is characteristic of neuronal degeneration in patients with and mouse models of polyglutamine disease (*Friedman et al., 2007*).

A molecular understanding of LCD is necessary to determine the prevalence and importance of this process in development. Genetic studies of *C. elegans* linker cell death have identified genes that promote this process, including *pqn-41*, which encodes a glutamine-rich protein of unknown

function, and *tir-1*/TIR-domain and *sek-1*/MAPKK, which may function in the same pathway (*Blum et al., 2012*). Intriguingly, the *Drosophila* and vertebrate homologs of TIR-1 promote distal axon degeneration following axotomy (*Osterloh et al., 2012*), supporting a conserved role for this protein in cell and process culling. The *let-7* microRNA and its indirect target, the Zn-finger transcription factor LIN-29, also promote LCD, and may act early in the process (*Abraham et al., 2007*; *Blum et al., 2012*). Nonetheless, the molecular logic of LCD is not understood.

Here, we describe a molecular-genetic framework governing LCD in *C. elegans*. Our studies represent the first such framework for a non-apoptotic cell death program regulating developmental physiology. We demonstrate that LCD is controlled by two Wnt signals, one pro-death and one pro-survival, that function in parallel, and partially redundantly with the LIN-29, and SEK-1/MAPKK pathways to control non-canonical activity of HSF-1, a conserved transcription factor that mediates heat-shock and other stress responses. Our functional, genetic, and molecular studies demonstrate that HSF-1 adopts a specific role, distinct from its well-described protective role in the heat-shock response, to promote LCD. We show that *let-70*, encoding a conserved E2 ubiquitin-conjugating enzyme, is an important transcriptional target of this pro-death developmental activity of HSF-1, but not of the HSF-1 stress-response function. LET-70 expression, as well as expression of ubiquitin and some proteasome components, increases just before LCD onset, and this increase requires the Wnt, LIN-29, SEK-1/MAPKK pathways, and HSF-1. CUL-3/cullin, RBX-1, BTBD-2, and SIAH-1 E3-ubiquitin ligase components function in the same pathway as LET-70 and promote LCD.

Our studies reveal design similarities between LCD and apoptosis. In *C. elegans*, cell lineage specifies the initiation of developmental apoptosis by transcriptional induction of the *egl-1* gene (*Thellmann et al., 2003*), encoding a pro-apoptotic BH3-only protein, or the *ced-3* gene, encoding the key executioner caspase (*Maurer et al., 2007*). Pathways linking cell lineage specification to transcriptional initiation of apoptosis have been described for some cells and appear to consist of multiple coordinated inputs. Thus, in both LCD and apoptosis diverse signals control specific transcriptional inputs that, in turn, control protein degradation machinery.

The molecular conservation of all the elements comprising the LCD program, together with the characteristic cell death ultrastructure, suggest that this program may be broadly conserved and provide an opportunity for probing the process in other settings.

## Results

### An EGL-20/Wnt pathway promotes linker cell death

To determine how LCD is initiated, we noted that mutations in the gene *him-4*, encoding the secreted protein hemicentin, prevent posterior migration of the linker cell, and result in low-level (~15%) linker cell survival (*Abraham et al., 2007*). Thus, linker cell position might, in part, dictate cell death onset. We considered the possibility that secreted ligands of the Wnt pathway, which are expressed in restricted spatial domains in *C. elegans*, contribute to LCD. We examined animals carrying lesions in each of the five *C. elegans* Wnt genes and found that in *egl-20*/Wnt mutants, the linker cell survives inappropriately (*Figure 1A,B*), and surviving cells are not engulfed (*Figure 1—figure supplement 1*). Importantly, linker cell migration, a complex multi-step process dependent on many genes (*Schwarz et al., 2012*), is unaffected in *egl-20* single mutants. Likewise, expression of reporter genes, including *lag-2* promoter::GFP (*Figure 1B*, *Figure 1—figure supplement 1A,B*), appears unaffected. Thus, *egl-20* mutations do not generally perturb linker cell fate, suggesting that the gene has a specific role in LCD control.

To determine whether EGL-20 promotes LCD in the context of Wnt signaling, we examined mutants defective in other pathway components. Animals carrying mutations in the *mig-14*/Wntless gene, which is required for Wnt secretion, also exhibit surviving linker cells at the cloaca (*Figure 1A*). Similarly, *mig-5*/Dishevelled and *bar-1*/β-catenin mutants, as well as *lin-17*/Frizzled; *mom-5*/Frizzled double mutants, exhibit linker cell survival without defects in migration or reporter expression (*Figure 1A*). Other Wnt mutants or mutant combinations do not block LCD (*Supplementary file 1A*). The kinase GSK3β curtails Wnt signaling by promoting degradation of β-catenin, and a *gsk-3* mutation restores LCD to *egl-20* mutants (*Figure 1A*). Furthermore, a heat-shock-inducible promoter driving a cDNA encoding a stabilized N-terminally-truncated BAR-1/β-catenin protein (*hsp-16.2* promoter::ΔN-*bar-1*) displays heat-shock-dependent restoration of LCD not only to *bar-1*/β-catenin

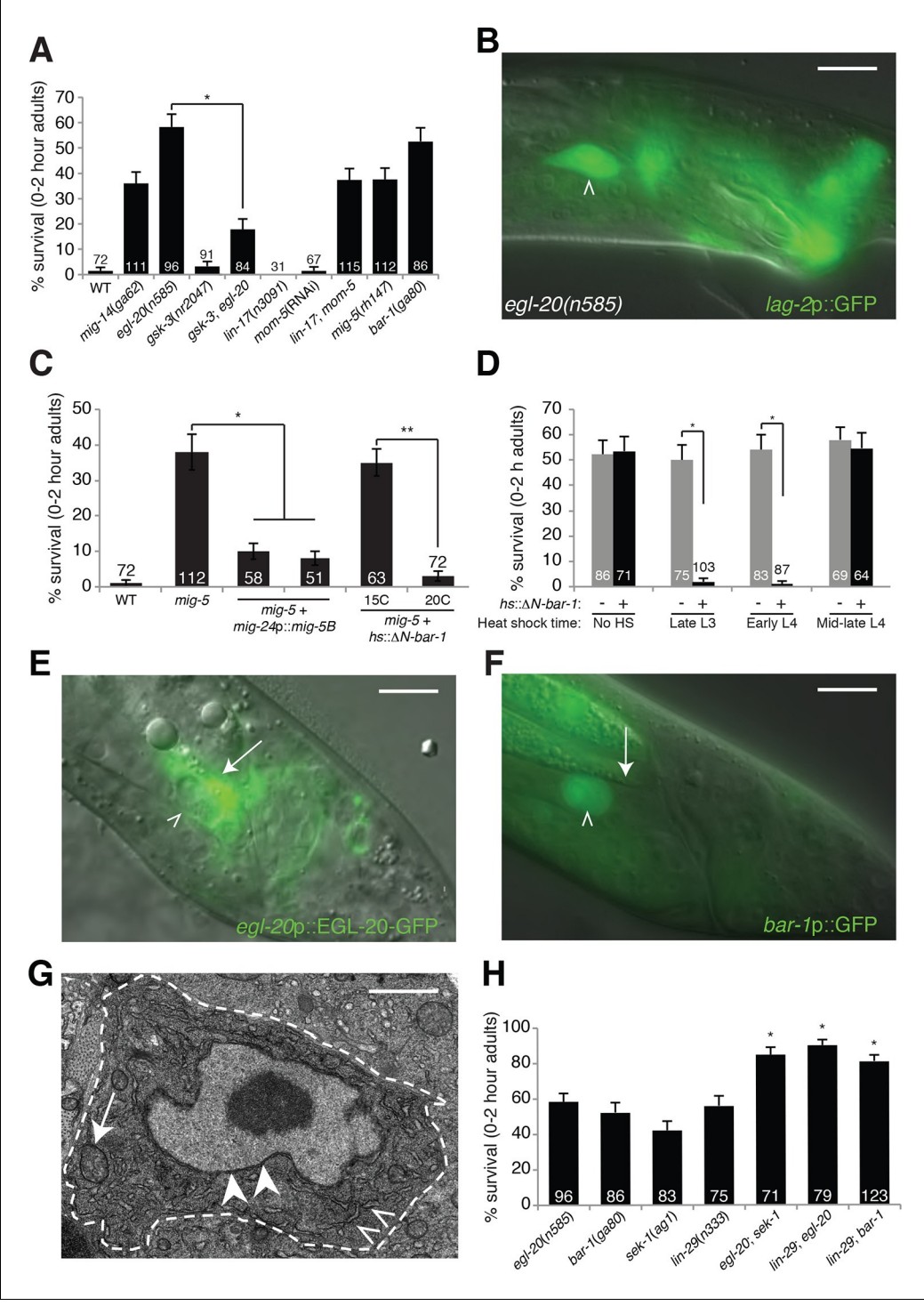

**Figure 1.** An *egl-20/*Wnt pathway promotes LInker cell death. (**A**) Linker cell survival in indicated genotypes. Strains contain *qIs56[lag-2*p::GFP] linker cell reporter transgene and *him-5(e1490)* for males. *gsk-3(nr2047)* is linked to *unc-101(sy216)*. *p<10$^{-4}$, no. animals scored is inside bars. (**B**) Adult *egl-20(n585)* male expressing *lag-2*p::GFP. (**C**) Linker cell survival in *mig-5(rh147)* animals with indicated transgenes. *p<10$^{-4}$, **p<.002. (**D**) *bar-1(ga80)* rescue with *hsp-16.2*p::ΔN-BAR-1. *p<10$^{-4}$. (**E**) *egl-20*p::EGL-20::GFP expression in L4 male. (**F**) *bar-1*p::GFP expression in L4 male. In (**B**), (**E**), (**F**), white caret, linker cell; arrow, UI/r.p cells; scale bar, 10 μm. (**G**) EM of surviving linker cell in *bar-1(ga80)* adult. Arrow, mitochondria. Arrowheads, nuclear envelope. Carets, healthy ER. Scale bar, 1 μm. (**H**) Linker cell survival in indicated genotypes. *p<10$^{-4}$ from the single mutant.

*Figure 1 continued on next page*

*Figure 1 continued*

The following figure supplements are available for figure 1:

**Figure supplement 1.** Surviving linker cells in *egl-20* mutants are not engulfed, but dying ones are.

**Figure supplement 2.** Expression of receptive Wnt components in the linker cell.

mutants, but also to *mig-5*/Dishevelled mutants (*Figure 1C,D*). These data support involvement of a canonical Wnt pathway in promoting LCD.

## EGL-20/Wnt pathway components function in the linker cell just before death

Cells surrounding the hermaphrodite cloaca have been previously shown to express EGL-20 (*Whangbo and Kenyon, 1999*). These cells, including the U.l/rp cells that engulf the linker cell, but not the linker cell, also express EGL-20 in males at the time of LCD (*Figure 1E*), consistent with a specific role in LCD.

To determine whether receptive Wnt components function in the linker cell to promote its demise, we examined their expression patterns. An 11-kb regulatory region upstream of the *bar-1/*β-catenin gene fused to GFP is not expressed in cloacal cells or in the trailing gonad but is strongly expressed in the linker cell (*Figure 1F*). Likewise, *mig-5*/Dishevelled::GFP and *lin-17*/Frizzled::GFP reporters are expressed in the linker cell (*Figure 1—figure supplement 2A,B*). Consistent with these data, expression of a *mig-5*/Dishevelled cDNA using a linker-cell-specific *mig-24* promoter (*Tamai and Nishiwaki, 2007*) restores cell death to *mig-5* mutant males (*Figure 1C*), indicating a cell-autonomous role for this gene.

To examine when Wnt signaling is required for LCD, we heat shocked *bar-1/*β-catenin mutants carrying a heat-inducible *hsp-16.2* promoter::ΔN-*bar-1* transgene at different time points during larval development, and assessed restoration of cell death. We found that induction as late as the early L4 stage rescued inappropriate linker cell survival (*Figure 1D*), suggesting that *bar-1* activity just before cell death onset is likely sufficient to drive cell death. This observation also supports the notion that EGL-20/Wnt signaling specifically controls LCD and not identity.

Unlike surviving cells in *pqn-41* or *sek-1* mutants, in which organelle changes accompanying cell death are evident (*Blum et al., 2012*), surviving linker cells in *bar-1/*β-catenin mutants do not exhibit death-associated ultrastructural features (*Abraham et al., 2007*) (*Figure 1G*), supporting a role for the Wnt pathway in cell death initiation. Taken together, our data suggest that the linker cell responds to an EGL-20/Wnt signal emanating from surrounding cells just prior to its death, using redundant activities of the receptors LIN-17 and MOM-5 and the signal transduction components MIG-5/Dishevelled and BAR-1/β-catenin.

Unexpectedly, mutations in *pop-1,* the sole *C. elegans* homolog of the canonical Wnt signaling transcription factor Tcf, cause no or weak defects in LCD (*Figure 2—figure supplement 1A*). Furthermore, while RNAi against *pop-1*/Tcf promotes highly penetrant defects in other contexts in *C. elegans* (*Siegfried and Kimble, 2002*), only low-level linker cell survival is evident even in RNAi-sensitized backgrounds (*Figure 2—figure supplement 1A*). *pop-1*/Tcf lesions also do not enhance or suppress linker cell survival in *egl-20*/Wnt mutants (*Figures 2A*, *Figure 2—figure supplement 1A*), and a *pop-1*/Tcf activity reporter is not expressed in the linker cell before or during death (*Figure 2—figure supplement 1B–D*). Likewise, while BAR-1/β-catenin physically and functionally interacts with the transcription factor DAF-16/FOXO (*Essers et al., 2005*), we found that a *daf-16* mutation does not block LCD (*Supplementary file 1A*).

## LIN-44/Wnt promotes linker cell survival

While testing genetic interactions between *egl-20*/Wnt mutants and other *C. elegans* Wnt mutants, we found, surprisingly, that mutations in *lin-44*/Wnt strongly suppressed inappropriate linker cell survival in *egl-20* mutants (*Figure 2A*). These data suggest that two opposing Wnt pathways control LCD: an EGL-20/Wnt pathway promotes, and a LIN-44/Wnt pathway prevents cell death. To test this idea, we examined genetic interactions between EGL-20/Wnt pathway components and other related genes. LCD is also restored to *egl-20*/Wnt mutants by mutations in *mig-1*/Frizzled, *cfz-2/*

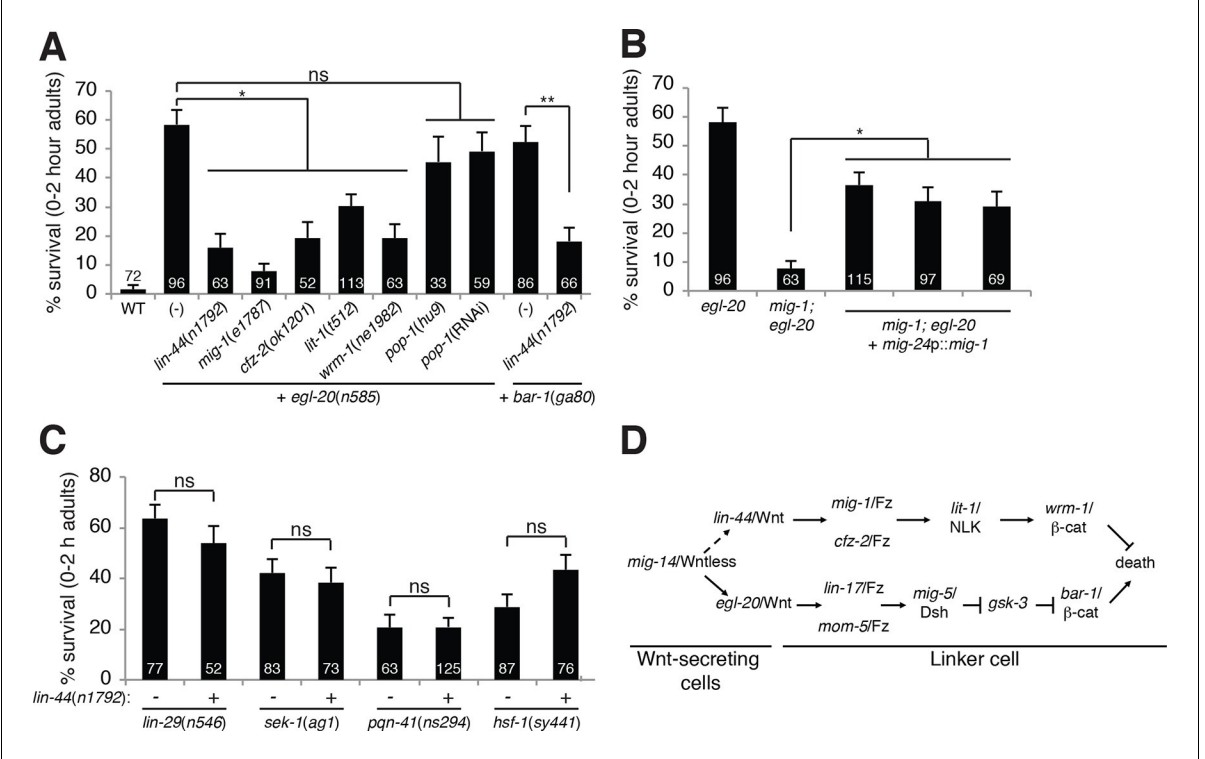

**Figure 2.** A *lin-44*/Wnt pathway promotes linker cell survival. (**A**) Linker cell survival in indicated genotypes. In (**A-C**) strains also contain *qIs56* and *him-5* (*e1490*). *p<10$^{-3}$; **p <10$^{-4}$; ns, not significant; Fisher's exact test. *lit-1(t512)* is linked to *unc-32(e189)*. (**B**) Linker cell survival in *egl-20(n585)* and *mig-1* (*e1787*); *egl-20(n585)* animals harboring a *mig-24*p::*mig-1* transgene. *p<0.001. (**C**) Linker cell survival in indicated genotypes. ns, not significant; Fisher's exact test. (**D**) Model for Wnt pathway interactions in LCD.

The following figure supplement is available for figure 2:

**Figure supplement 1.** *pop-1* does not play a significant role in linker cell death.

Frizzled, *lit-1*/NLK or *wrm-1*/β-catenin. *lin-44*/Wnt mutations also suppress inappropriate linker cell survival in *bar-1* mutants (**Figure 2A**).

*lin-44* is expressed in the *C. elegans* male tail (**Figure 1—figure supplement 2C**) (**Herman et al., 1995**), consistent with a role in LCD. *wrm-1*/β-catenin is expressed in the linker cell, as well as other cells (**Figure 1—figure supplement 2D**). Furthermore, expression of a *mig-1*/Frizzled cDNA specifically in the linker cell restores inappropriate linker cell survival to *mig-1*/Frizzled; *egl-20*/Wnt double mutants (**Figure 2B**). These results suggest that a tail-derived LIN-44/Wnt signal impinges on the MIG-1/Frizzled and CFZ-2/Frizzled receptors. These receptors function together in the linker cell, through *lit-1*/NLK and *wrm-1*/β-catenin, to promote its survival (**Figure 2D**). While we were unable to score *wrm-1; bar-1* double mutants, as these have a fully penetrant, early block in linker cell migration (100%, n>100) as well as other defects in larval development, our results suggest that the EGL-20/Wnt pathway antagonizes the LIN-44/Wnt pathway at or downstream of WRM-1/β-catenin.

## EGL-20/Wnt and LIN-44/Wnt function in parallel to known LCD regulators

Null alleles of *egl-20*/Wnt block LCD in about 60% of animals (**Figure 1A**), and early expression of ΔN-BAR-1/β-catenin fails to promote premature onset of LCD (**Figure 1D**), suggesting that additional cues promote LCD initiation. The linker cell dies at a specific place and time during *C. elegans* male development, and previous studies showed that a developmental timing cue transduced by the Zn-finger transcription factor LIN-29 partially controls LCD (**Abraham et al., 2007**) (**Figure 1H**). In *lin-29; egl-20*/Wnt and *lin-29; bar-1*/β-catenin double mutants, nearly all linker cells survive inappropriately (**Figure 1H**), suggesting that the LIN-29 timing cue and the EGL-20/Wnt cue function in

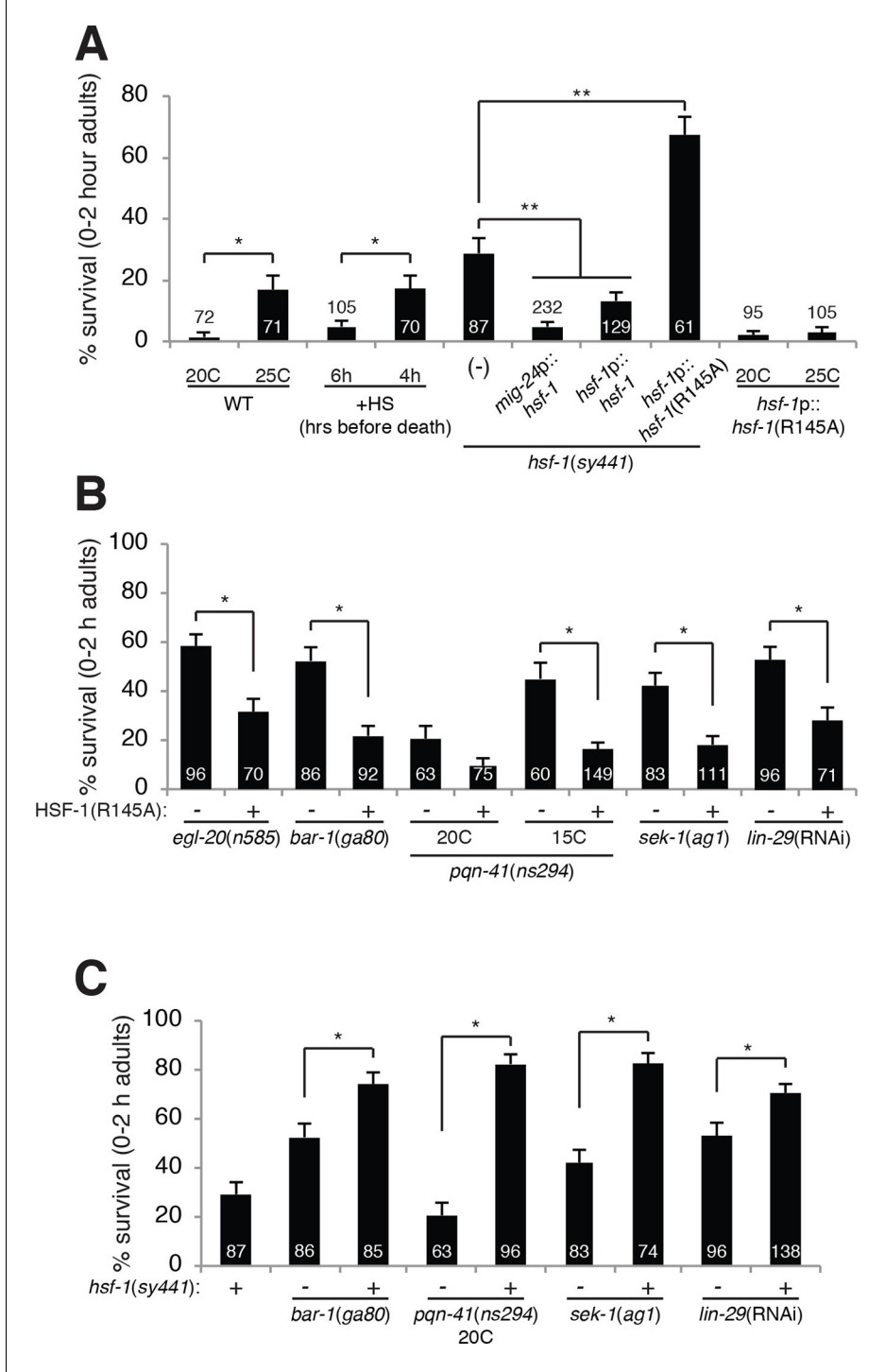

**Figure 3.** HSF-1 promotes linker cell death. (**A**) Linker cell survival in indicated genotypes. In (**A-C**), strains also contain *qIs56* and *him-5(e1490)*. *p<10⁻²;**p<10⁻³; Fisher's exact test. *hsf-1*p::*hsf-1*(WT/R145A) transgenes are fused to GFP. WT: animals raised at indicated temperature after hatching. +HS: WT animals heat shocked at 37°C for 15 min at 6 hr or 4 hr before the L4-adult molt. *hsf-1(sy441)*: *mig-24*p::*hsf-1* bar is average of three independent extrachromosomal array lines. *hsf-1*p::*hsf-1*(R145A) bar is average of two independent single-copy integrated lines. *hsf-1*p::*hsf-1*(R145A): animals were raised at the indicated temperature after hatching. (**B**) Linker cell survival in indicated genotypes. HSF-1(R145), *hsf-1*p::*hsf-1*(R145A). The *drSi28[hsf-1*p::*hsf-1*(R145A)]* transgene was used. For *hsf-1*p::*hsf-1*(R145A); *bar-1(ga80)*, two other independent single-copy integrated lines gave similar results. (**C**) Linker cell survival in indicated genotypes.

*Figure 3 continued on next page*

*Figure 3 continued*

The following figure supplement is available for figure 3:

**Figure supplement 1.** Markers of the heat-shock response are not induced during linker cell death.

---

parallel to control LCD initiation. We previously showed that the MAPKK gene *sek-1* also functions in parallel to *lin-29* (*Blum et al., 2012*). In *egl-20; sek-1* double mutants, nearly all linker cells also survive (*Figure 1H*). Furthermore, although *lin-44*/Wnt mutations suppress ectopic linker cell survival in *egl-20*/Wnt and *bar-1*/β-catenin mutants, they do not restore LCD to *lin-29, sek-1,* or *pqn-41* mutants (*Figure 2C*). Thus, EGL-20/Wnt and LIN-44/Wnt, LIN-29, and SEK-1/MAPKK define three parallel, partially redundant pathways initiating LCD.

## HSF-1 promotes LCD independently of the heat-shock response

Heat-shock factors are transcriptional regulators, activated in response to certain stresses including heat shock, whose targets include chaperones and other effectors that maintain cell viability during stress. While exploring roles for stress response genes in LCD, we found that a hypomorphic allele (*sy441*) of the single *C. elegans* heat-shock factor gene, *hsf-1*, causes inappropriate linker cell survival (*Figure 3A*). The *hsf-1(sy441)* allele is a loss-of-function allele that truncates the region encoding the HSF-1 transcriptional transactivation domain (*Hajdu-Cronin et al., 2004*). This observation suggests, surprisingly, that rather than protecting the linker cell, HSF-1 promotes its demise. Indeed, either a single copy *hsf-1* promoter::*hsf-1*::GFP transgene, expressed at roughly the same level as the native *hsf-1* locus (*Morton and Lamitina, 2013*), or a wild-type *hsf-1* cDNA expressed specifically in the linker cell using the *mig-24* promoter, rescue the *hsf-1(sy441)* LCD defect (*Figure 3A*), showing that *hsf-1* can function cell-autonomously to promote LCD.

Compromised stress responses do not generally block LCD, as neither unfolded protein response mutants (*Blum et al., 2012*), nor *daf-16*/FOXO or *daf-21*/HSP90 mutants (*Supplementary file 1A*) show LCD defects (*Blum et al., 2012*). This raises the possibility that the role of HSF-1 in LCD may be different from its role in the canonical heat-shock response. To test this idea directly, we examined expression of GFP reporters for HSF-1 target genes normally induced during heat shock and found that they are not induced in the linker cell during LCD (*Figure 3—figure supplement 1A–D*). Supporting this conclusion, the *hsp-16.2* gene is a target of HSF-1 in the heat-shock response, and LCD is restored to *bar-1* mutants by the *hsp-16.2* promoter::ΔN-*bar-1* transgene following a heat shock. However, no rescue is evident without heat exposure (*Figure 1D*), suggesting that the *hsp-16.2* promoter is not normally induced during LCD. Similarly, while nuclear-cytoplasmic shuttling does not control HSF-1 activity in *C. elegans* (*Morton and Lamitina, 2013*), HSF-1 does form nuclear aggregates in all cells following stress exposure (*Morton and Lamitina, 2013*). While aggregates can be seen in dying linker cells in stressed animals (*Figure 3—figure supplement 1E*), no aggregates are evident in the dying linker cell, or the surrounding cells, of unstressed animals (*Figure 3—figure supplement 1F*), supporting a novel role for HSF-1 in LCD.

In addition to functional differences between the role of HSF-1 in the heat-shock response and LCD, we also found distinct genetic requirements. The HSF-1(R145A) protein contains a mutation in the putative HSF-1 DNA binding domain. Previous studies demonstrated that expression of this protein restores the heat-shock response to *hsf-1(sy441)* mutants lacking the distal portion of the HSF-1 transactivation domain (*Morton and Lamitina, 2013*). Trans-complementation in the active HSF-1 trimer likely explains how these two loss-of-function lesions can, together, promote a normal heat-shock response. However, instead of rescuing the LCD defect of *hsf-1(sy441)* mutants, we found that a single copy *hsf-1*(R145A) transgene enhanced inappropriate linker cell survival from 29% to 61% (*Figure 3A*).

Taken together, our results show that the role of HSF-1 in LCD is functionally and genetically distinct from its role in the heat-shock response. A prediction arising from these data is that the LCD and heat-shock functions of HSF-1 might compete with each other. To test this, we first observed that while the linker cell of wild-type males raised at 20°C always dies, some wild-type adult males raised at 25°C harbor a surviving linker cell (*Figure 3A*), suggesting that the heat-shock role of HSF-1 might compete with its LCD role. To test this more directly, we subjected males to a 37°C heat

shock 4 hr prior to LCD onset and found that these animals also exhibit a surviving linker cell. Importantly, males heat-shocked 6 hr before LCD onset exhibit fewer surviving linker cells (*Figure 3A*). These results are consistent with the idea that heat-shock functionally sequesters HSF-1 away from its LCD role, and that activity of HSF-1 just before the cell dies is required to promote LCD. These results also explain why we were able to rescue *bar-1* mutants with the *hsp-16.2* promoter::ΔN-*bar-1* transgene, as heat shock was performed 10 hr before LCD onset, well before the activity of HSF-1 is required.

## HSF-1 promotes death downstream of known LCD regulators

An examination of males carrying the *hsf-1*(R145A) transgene in an otherwise wild-type background revealed that LCD progressed to completion in all animals even at 25°C (*Figure 3A*). This result suggests that in a wild-type *hsf-1* background, *hsf-1*(R145A) functions as a gain-of-function allele, promoting LCD. One possibility for how this might occur is that the allele preferentially disrupts HSF-1 complexes promoting the heat-shock response, thereby promoting LCD. Regardless of the precise mode of action, our serendipitous discovery of the gain-of-function nature of the R145A protein allowed us to dissect the functional relationships between HSF-1 and the parallel pathways controlling LCD onset. Strikingly, we found that three independent *hsf-1*(R145A) single-copy transgene isolates restored LCD to *egl-20*/Wnt and *bar-1*/β-catenin mutants (*Figure 3B*). Importantly, a *lin-44/* Wnt mutation could not restore cell death to *hsf-1(sy441)* animals (*Figure 2C*). Likewise, *hsf-1* (R145A) transgenes also restored LCD to *lin-29, sek-1*/MAPKK, and *pqn-41*/Q-rich mutants (*Figure 3B*). These results suggest that the Wnt, LIN-29, and SEK-1/PQN-41 pathways all require HSF-1 function to promote LCD.

Consistent with these observations, we found a synergistic increase in linker cell survival in animals carrying mutations in *egl-20, lin-29, sek-1*, or *pqn-41* and the *hsf-1(sy441)* partial loss-of-function mutation (*Figure 3C*), as might be predicted if HSF-1 functions downstream of all LCD initiation pathways we described.

## LET-70/UBE2D2, an E2 ubiquitin-conjugating enzyme, is required for linker cell death

To understand the mechanism by which HSF-1 promotes LCD, we sought genes that function downstream. We previously performed a genome-wide RNA interference (RNAi) screen identifying genes required for LCD (described in *Blum et al., 2012*). From this screen, we found that males fed bacteria expressing dsRNA targeted against the gene *let-70*, encoding a putative E2 ubiquitin-conjugating enzyme, exhibit robust linker cell survival, indicating that the gene is required for LCD (*Figure 4A-C, 4E*). As RNAi can induce off-target effects, we confirmed our results by examining two non-overlapping RNAi targeting fragments and obtained similar results (*Figure 4A,E*). *let-70*(RNAi) animals exhibit surviving linker cells with normal ultrastructure (*Figure 4C*), indicating that *let-70* likely acts in promoting LCD and not in corpse degradation. Consistent with this observation, surviving linker cells are unengulfed (*Figure 4—figure supplement 1A*).

To confirm the *let-70* RNAi results, we sought animals carrying inactivating mutations in the gene. Animals homozygous for a previously isolated allele, *tm2777*, or a CRISPR/Cas9-induced deletion we generated, *ns636*, exhibit larval lethality and adult sterility as previously reported for the *s689* allele (*Zhen et al., 1993*), precluding studies of LCD. However, *ns770*, a CRISPR/Cas9-induced C-to-T point mutation we made that is predicted to generate a P61S alteration in the LET-70 protein, is viable (*Figure 4A*). A similar lesion confers instability to the *S. cerevisiae* UBC4 E2 enzyme at 39°C (*Tongaonkar et al., 1999*), suggesting that *ns770* may be a partial loss-of-function allele. Indeed, we found that 57% of *let-70(ns770)* animals possess surviving linker cells (*Figure 4E*). This defect is not temperature dependent in the growth range of *C. elegans* (15°C: 62%, n=78; 25°C: 57%, n=87), perhaps because these temperatures are much lower than those abolishing function in yeast.

To determine where LET-70 acts to promote death, we generated *let-70(ns770)* animals carrying a *mig-24* promoter::*let-70* cDNA construct expressed specifically in the linker cell. As shown in *Figure 4E*, LCD is restored in these animals, suggesting that *let-70* acts within the linker cell to promote its demise (*Figure 4E*). To confirm this idea, we carried out linker-cell-specific *let-70* RNAi. RDE-1 is an argonaute protein required for RNAi, and in *rde-1(ne219); mig-24* promoter::*rde-1* cDNA animals, RNAi is only functional in the linker cell (*Figure 4—figure supplement 1B*). RNAi

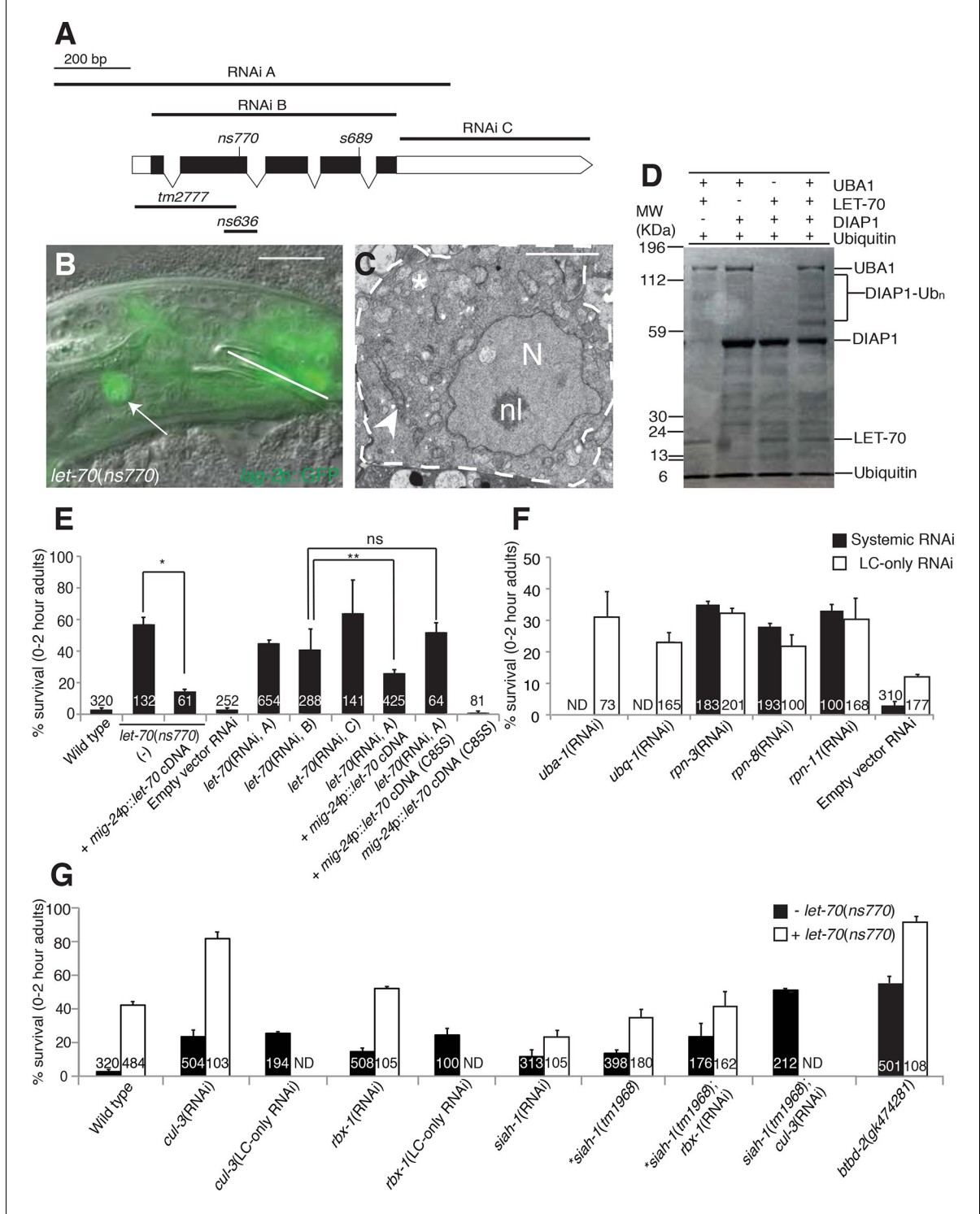

**Figure 4.** *let-70* promotes linker cell death. (**A**) *let-70* gene structure and mutations/RNAi clones used in our studies. Black boxes, exons; white boxes, 5' or 3' untranslated regions. Scale bar, 200 bp. (**B**) Combined DIC and fluorescent images of *let-70*(RNAi) adult male. *lag-2*p::GFP marks the linker cell. Arrow, linker cell. White line, cloaca. Scale bar, 10 μm. (**C**) EM of surviving *let-70*(RNAi) linker cell. Scale bar, 2 μm. Asterisk, mitochondria, Arrowhead, ER, N, nucleus, nl, nucleolus. (**D**) Purified 6xHis-LET-70, *Drosophila* UBA1, DIAP1 and ubiquitin. causes DIAP1 auto-ubiquitination. (**E-H**) Linker cell survival in indicated genotypes. No. animals scored, inside bars. Error bars, SEM. *p<0.001; **p<0.0001; Fisher's Exact Test; ns, not significant. Animals contained *qIs56* and *him-5(e1490)*. In (**F**) animals also contained *rrf-3(pk1426)* for increased RNAi efficiency. In LC-only experiments, *mig-24*p was used to drive *rde-1* cDNA in *rde-1(ne219)*; *him-8(e1489)*; *qIs56* mutants. ND, not determined.

*Figure 4 continued on next page*

*Figure 4 continued*

The following figure supplements are available for figure 4:

**Figure supplement 1.** *let-70*(RNAi) animals have unengulfed linker cells.
**Figure supplement 2.** Expression of *siah-1, rbx-1*, and *cul-3* in migrating and dying linker cells.

against *let-70* in these animals also prevents LCD (*Figure 4E*). We conclude that LET-70 acts cell autonomously to kill the linker cell.

*let-70* is predicted to encode a protein 94% identical to the mammalian E2 ubiquitin-conjugating enzyme UBE2D2 (*Zhen et al., 1993*). To confirm that LET-70 indeed functions as an E2 enzyme, we assayed its ability to mediate ubiquitin transfer in an in vitro ubiquitylation assay. Incubation of LET-70 with the *Drosophila* UBA1 E1-activating enzyme and the *Drosophila* E3 ligase DIAP1 results in DIAP autoubiquitylation in the presence of ATP, magnesium ions, and ubiquitin (*Figure 4D*). A similar reaction without LET-70 does not yield DIAP ubiquitylation, suggesting that LET-70 functions as an E2. To determine whether LET-70 functions as an E2 enzyme in vivo, we first examined *let-70 (ns770)* animals expressing a *mig-24* promoter::*let-70*(C85S) cDNA transgene predicted to change the LET-70 catalytic cysteine 85 to serine. However, transgenic animals exhibited hermaphrodite sterility (presumably due to expression in the hermaphrodite distal tip cell required for gonad development), suggesting that LET-70(C85S) is a dominant negative protein. However, in another set of experiments, we found that while a silently-mutated RNAi-resistant *let-70* cDNA partially rescues the LCD defect of *let-70*(RNAi) animals, a similar cDNA encoding the C85S mutation does not (*Figure 4E*). Taken together, our studies suggest that the ubiquitin-conjugating activity of LET-70 is required in vivo for LCD.

To determine whether other E2 enzymes are also required for LCD, we performed RNAi against 13/22 E2-encoding genes with available dsRNA bacterial clones and found no evidence of inappropriate linker cell survival, indicating that LET-70 likely acts specifically to promote LCD (*Supplementary file 1B*).

## The proteasome and other UPS components promote linker cell death

To determine whether LET-70 functions as part of the ubiquitin proteasome system (UPS) for LCD, we first tested if UBA-1, the sole E1 activating enzyme in *C. elegans*, is also required. While systemic RNAi against *uba-1* is early-larval lethal, linker-cell-specific RNAi against *uba-1* produces a robust defect in LCD (*Figure 4F*). Similarly, weak *uba-1(it129)* mutant animals survive to adulthood and display weak but significant linker cell survival (17 ± 2% survival, n=209). Furthermore, RNAi against the gene encoding ubiquitin, *ubq-1*, also blocks LCD (*Figure 4F*). Thus, LCD requires canonical components of the ubiquitin-mediated protein degradation pathway.

We also examined the effects of inhibiting components of the 19S proteasome regulatory subunit on LCD and found that systemic RNAi against the *rpn-3, rpn-8*, or *rpn-11* genes results in linker cell survival in about one third of animals (*Figure 4F*), and linker cell-specific RNAi against these genes results in similar inhibition (*Figure 4F*). Taken together, our results strongly suggest that LET-70 functions in the linker cell as a component of the UPS.

## The E3 components CUL-3, RBX-1, BTBD-2, and SIAH-1, function with LET-70 to promote linker cell death

E2 enzymes such as LET-70 function through E3 proteins to mediate protein degradation (*Hershko et al., 1983*). We therefore sought to identify E3 ubiquitin ligase components that mediate LET-70 activity. Cullin proteins are subunits of many E3 enzymes, and the *C. elegans* genome encodes six such proteins (CUL-1-6). We tested whether any of these is involved in LCD and found that RNAi against the *cul-3* gene results in inappropriate linker cell survival (*Figure 4G*, *Supplementary file 1C*). Linker-cell-specific RNAi against *cul-3* also yields surviving linker cells, supporting a cell autonomous function for this gene. Strikingly, *cul-3*(RNAi); *let-70(ns770)* animals exhibit a synergistic increase in linker cell survival well above each single mutant, indicating that these genes likely function together, in sequence or in parallel, to promote LCD (*Figure 4G*).

Previous studies had demonstrated interactions between CUL-3 and the RING protein RBX-1 in *C. elegans* (*Pintard et al., 2003*). While many RING-finger encoding genes we tested by RNAi do not appear to have roles in LCD (*Supplementary file 1B*), we found that RNAi against the *rbx-1* gene does promote modest linker cell survival (*Figure 4G*), suggesting a role in LCD. Supporting this notion, *cul-3* and *rbx-1* are both expressed in the linker cell (*Figure 4—figure supplement 2*).

CUL-3 E3 complexes often contain BTB-domain substrate binding proteins. We screened 23 BTB proteins by RNAi and/or mutation, and identified two that block LCD when inactivated (*Supplementary file 1C*). One of these, EOR-1, will be described elsewhere. The other, BTBD-2, is homologous to human BTBD2, and its inactivation results in linker cell survival in roughly half of animals examined (*Figure 4G*). Expression of BTBD-2 using the *mig-24* linker-cell-specific promoter restored linker cell death to *btbd-2(gk474281)* mutants (47 ± 3% survival in *btbd-2(gk474281)* mutants (N=90) vs. 31 ± 4% survival in transgenic lines, 2 lines examined (N=81)).

We also examined 55 genes encoding protein domains commonly found in E3 enzymes (*Supplementary file 1B,C*). We found that RNAi against the seven-in-absentia homolog *siah-1* causes a modest but significant linker cell survival defect (*Figure 4G*). To confirm this observation, we examined animals defective for the *siah-1(tm1968)* mutation, which deletes most of exon 4 of the gene and is likely a molecular null, and found a similar survival defect. Interestingly, both *siah-1(tm1968); cul-3*(RNAi) and *siah-1(tm1968); rbx-1*(RNAi) double mutants exhibit greater linker cell survival than either single mutant (*Figure 4G*). We conclude that CUL-3, RBX-1, BTBD-2, and SIAH-1, all function to promote LCD and likely do so downstream of LET-70.

## LET-70 expression is induced at the time of linker cell death and requires known linker cell death genes

To study the expression and localization of LET-70, we generated animals carrying *let-70* promoter:: *let-70*::GFP or *let-70* promoter::GFP transgenes. We found that both reporters are expressed in the linker cell and that the translational fusion reporter is evenly distributed between the nucleus and cytoplasm (*Figure 5B*, data not shown). Importantly, we found that expression of neither reporter is constitutive. Rather, while GFP fluorescence is not detected during migration of the linker cell, it is induced 1–2 hr before obvious morphological features of cell death appear (*Figure 5A–C*). Similar induction is seen with a fosmid recombineered to contain 18.9 kb surrounding the genomic *let-70* locus fused to GFP (n>25). We wondered whether the expression of other components of the UPS might also be induced in the linker cell with similar kinetics. Although some reporter genes we tested are not induced (*Figure 4—figure supplement 2*), expression of GFP reporter fusions to the *ubq-1* gene, encoding *C. elegans* ubiquitin, and to the proteasome component gene *rpn-3* is induced (*Figures 5D–I*). These results suggest that expression of some UPS components is upregulated in the linker cell just prior to cell death onset.

To understand how the induction of UPS genes is regulated, we looked at the expression of *let-70* promoter::*let-70*::GFP and *ubq-1* promoter::*ubq-1*::GFP reporter transgenes in surviving cells in mutants of the Wnt, LIN-29, and SEK-1/MAPKK pathways we identified as LCD regulators. Wild-type *rpn-3* promoter::GFP expression decreases in all cells in the first hours of adulthood and was not bright enough to reliably score in mutant backgrounds. We found that surviving linker cells in mutants of all three pathways often failed to express either GFP reporter (*Figure 5J*), but the effects were more pronounced for the *let-70* reporter. Double mutants between mutant components of each of the three regulatory pathways and *let-70(ns770* or RNAi) demonstrate additive interactions (*Figure 5K*), as would be expected with combinations of partial loss-of-function mutants functioning in the same pathway. Taken together, our results are consistent with a model in which LET-70 functions downstream of the Wnt, LIN-29, and SEK-1/MAPKK signals.

## LET-70 functions downstream of HSF-1

To examine the relationship between *let-70* and *hsf-1*, we looked at the expression of the *let-70* promoter::*let-70*::GFP and *ubq-1* promoter::*ubq-1*::GFP reporter transgenes in an *hsf-1(sy441)* partial loss-of-function mutant. As shown in *Figure 6A*, GFP expression was significantly reduced for both, with a more pronounced effect for the *let-70* reporter. These studies indicate that wild-type HSF-1 activity is required to induce *let-70* expression, and, therefore, that LET-70 functions downstream of HSF-1. *let-70* promoter::*let-70*::GFP expression is not induced by heat shock, consistent with

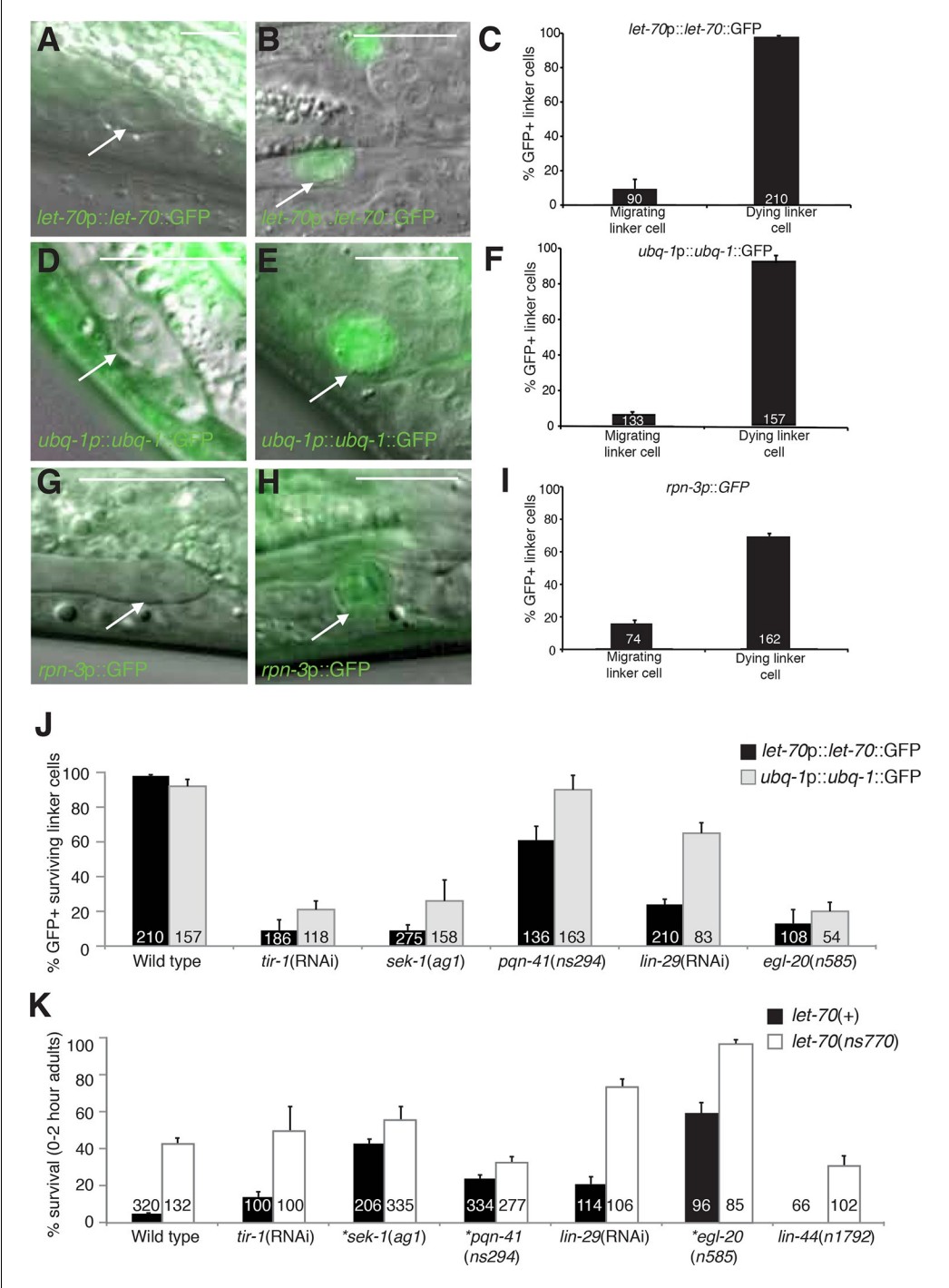

**Figure 5.** *let-70, ubq-1*, and *rpn-3* expression is induced just before linker cell death onset. (**A-C**) *let-70*p::*let-70*::GFP expression in migrating (**A**) or dying (**B**) linker cell. Scale bar, 10 μm. (**C**) Expression quantification in (**A,B**). Error bars, SEM. Number inside bar, no. animals scored. (**D-F**) Same as (**A-C**) for *ubq-1*p::*ubq-1*::GFP. (**G-I**) Same as (**A-C**) for *rpn-3*p::GFP. (**J**) Expression of indicated GFP reporters in surviving linker cells in *him-8(e1489)* animals of indicated genotype. (**K**) All animals contained *qIs56* and *him-5(e1490)*. *\*let-70*(RNAi) instead of *let-70(ns770)*.

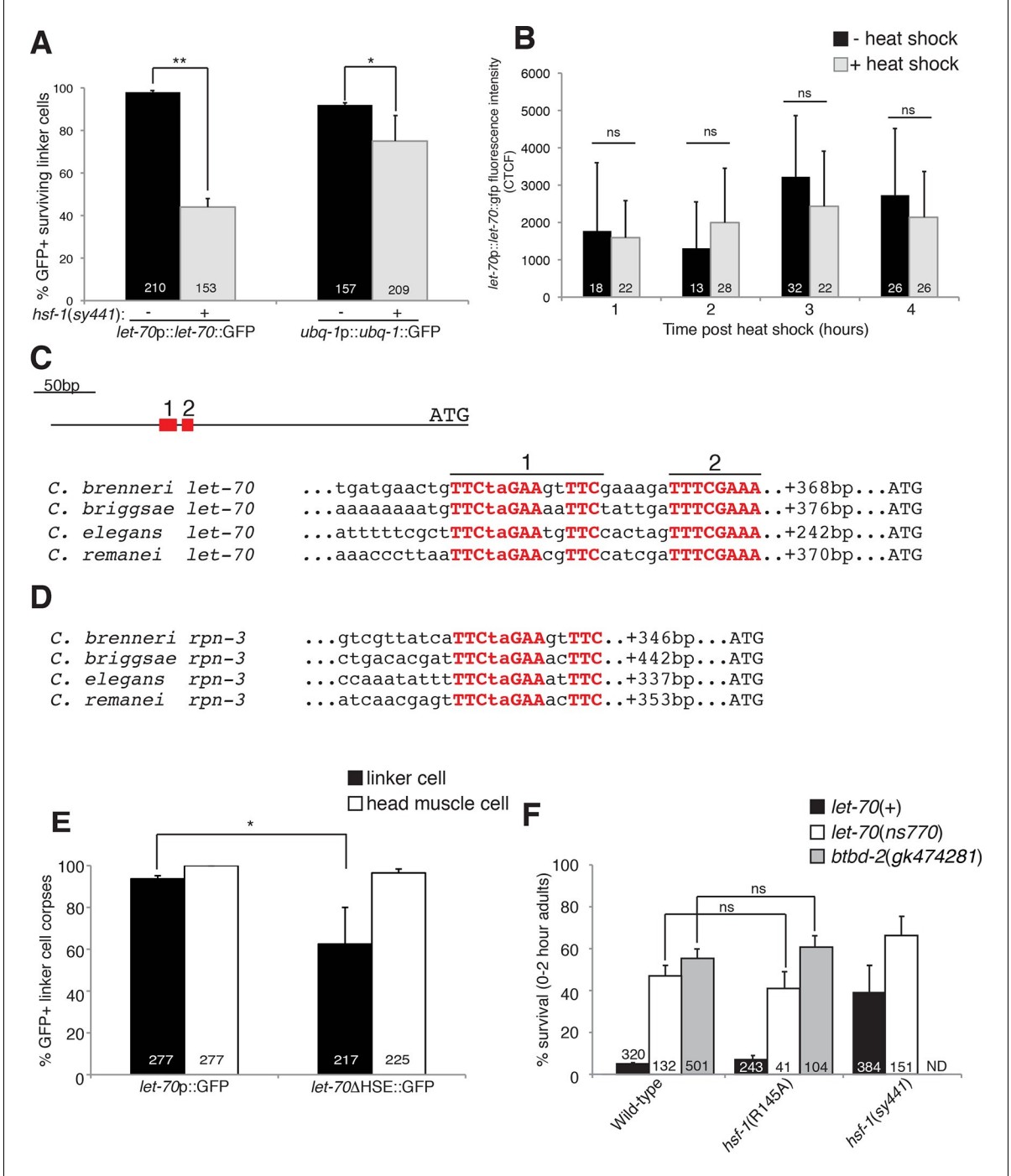

**Figure 6.** HSF-1 controls LET-70 expression. (**A**) Expression of indicated GFP reporter in surviving linker cells in *him-8(e1489)* animals of indicated genotype. **\*\*p<0.0001, \*p<0.005, Fisher's exact test. Error bars, SEM. (**B**) *let-70*p::*let-70*::gfp expression in head region after heat shock. Error bars, SD. ns, not significant, Student's t-test. (**C**) *let-70* promoter sequence alignment across indicated nematodes. Red, conserved nucleotides. (**D**) Same as (**C**) but for *rpn-3*. (**E**) *let-70*p::GFP and *let-70*ΔHSE::GFP expression. Error bar, SEM. \*p<0.0001, Fisher's exact test. (**F**) *let-70* and *btbd-2* interactions with *hsf-1*. Error bars, SEM. Number within bars, no. of animals scored. Animals contained *qIs56* and *him-5(e1490)*. ND= not determined.

The following figure supplement is available for figure 6:

**Figure supplement 1.** ΔHSE reduces *let-70* promoter::*let-70*::GFP expression in the linker cell.

previous Northern blot studies (*Figure 6B*) (*Zhen et al., 1993*). Therefore, consistent with our characterization of HSF-1, HSF-1 must be acting in a manner distinct from the heat-shock response to induce *let-70* expression and cell death in the linker cell.

The DNA motif TTCTAGAA is enriched in regulatory regions of genes induced in *C. elegans* in response to heat shock (*GuhaThakurta et al., 2002*), and the motif TTCnnGAAnnTTC has been defined as an HSF binding element from yeast to mammals. A comparison of *let-70* genomic sequences upstream of the ATG start codon revealed a region highly conserved between *C. elegans* and at least three other related nematode species (*Figure 6C*). Within this region we identified two conserved motifs. The upstream motif (motif 1) is identical to the HSF consensus binding site, whereas the downstream motif (motif 2) contains two potential HSF monomer binding sites (TTC and GAA). We also identified a highly conserved heat-shock element (HSE) upstream of the *rpn-3* gene (*Figures 5G-I*, *6D*). A consensus HSF binding site was not found within the regulatory sequences used for the *ubq-1* reporter studies; however, a number of one-off sites were found, perhaps explaining the weaker regulation of our *ubq-1* reporter by HSF-1.

To test the functional relevance of the *let-70* heat-shock element homology region for *let-70* expression, we generated animals harboring a *let-70* promoter::GFP reporter transgene in which a 97 nucleotide region including conserved motif 1 and 2 were deleted (*let-70ΔHSE::GFP*). As shown in *Figure 6E*, transgenic animals failed to express the reporter in the dying linker cell in about 40% of animals, consistent with the similar defect we observed in *let-70::GFP* expression in *hsf-1(sy441)* loss-of-function mutants. Importantly, GFP expression in other cells of *let-70ΔHSE::GFP* animals was not perturbed (*Figure 6E*, *Figure 6—figure supplement 1*), suggesting a specific role for this DNA element in controlling linker cell expression of *let-70*.

Our results demonstrate that *let-70* expression is under the control of HSF-1, likely acting through a consensus heat-shock element in the *let-70* 5' control region, but not through the canonical heat-shock response pathway. To functionally probe this model, we tested the genetic relationship

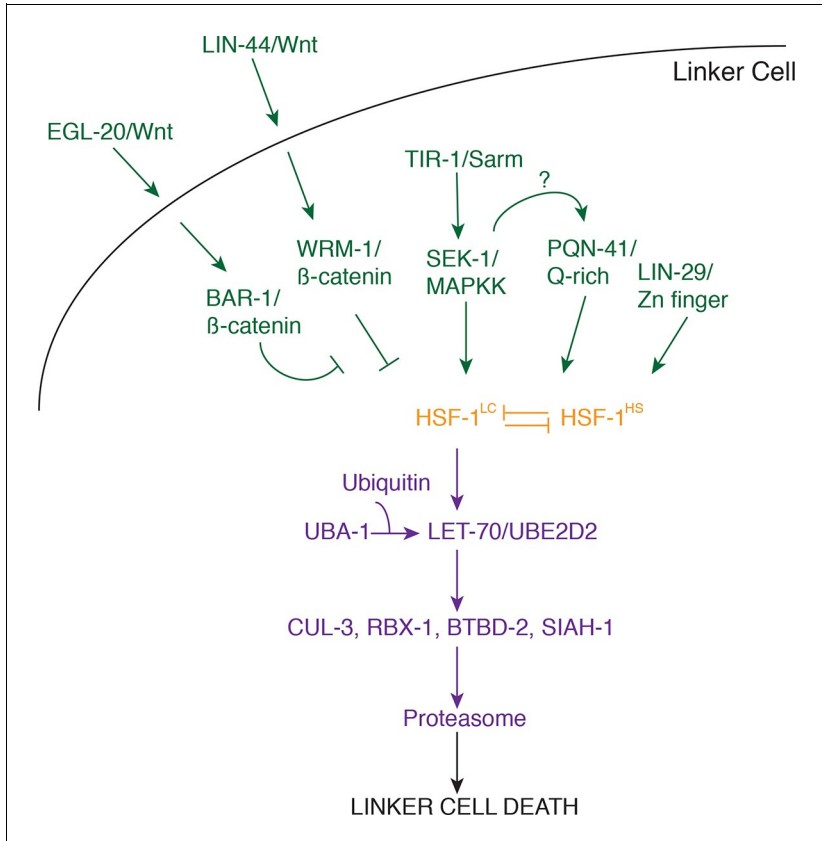

**Figure 7.** Model for linker cell death. Green, upstream regulators. Orange, HSF-1. Purple, proteolytic components.

between *let-70* and *hsf-1*. *hsf-1(sy441); let-70(ns770)* animals harboring partial loss-of-function perturbations of each gene, have increased linker survival compared to either single mutant alone (*Figure 6F*). More importantly, while the *hsf-1*(R145A) gain-of-function transgene restores cell death to all previously tested LCD mutants (see above), it fails to rescue inappropriate linker cell survival in *let-70(ns770)* mutants (*Figure 6F*). Likewise, the *hsf-1*(R145A) gain-of-function transgene fails to restore cell death to *btbd-2(gk474281)* mutants (*Figure 6F*).

Taken together, our results suggest that LET-70 and BTBD-2 function downstream of a linker-cell-specific non-canonical function of HSF-1 to promote LCD. Our data also suggest that other HSF-1 targets are likely relevant, and that *let-70* may be under the control of additional regulators.

## Discussion

### A new pathway for non-apoptotic cell death

The results presented here allow us to construct a model for the initiation and execution of LCD in *C. elegans* (*Figure 7*). The logic of the LCD pathway may be similar to that of developmental apoptotic pathways. In *C. elegans* and *Drosophila*, where the control of specific cell deaths has been primarily examined, cell lineage or fate determinants control the expression of specific transcription factors that then impinge on proteins regulating caspase activation (*Fuchs and Steller, 2011*). Likewise, LCD is initiated by redundant determinants that require a transcription factor to activate protein degradation genes.

Our data suggest that three partially redundant signals control LCD initiation. The antagonistic Wnt pathways we describe may provide positional information to the linker cell, as the relevant ligands are expressed only near the region where the linker cell dies. The LIN-29 pathway, which controls timing decisions during the L4-adult molt, may ensure that LCD takes place only at the right time. Finally, while the TIR-1/SEK-1 pathway could act constitutively in the linker cell, it may also respond to specific cues from neighboring cells. Indeed, MAPK pathways are often induced by extracellular ligands. We propose that these three pathways, together, trigger activation of HSF-1. Our data support a model in which HSF-1 is present in two forms, HSF-1$^{LC}$, promoting LCD, and HSF-1$^{HS}$, protecting cells from stresses, including heat shock. We postulate that the redundant LCD initiation pathways tip the balance in favor of HSF-1$^{LC}$, allowing this activity to bind to promoters and induce transcription of key LCD effectors, including LET-70/UBE2D2 and other components of the ubiquitin proteasome system (UPS), functioning through E3 ligase complexes consisting of CUL-3, RBX-1, BTBD-2, and SIAH-1.

Importantly, the molecular identification of LCD components and their interactions opens the door to testing the impact of this cell death pathway on vertebrate development. For example, monitoring UBE2D2 expression during development could reveal upregulation in dying cells. Likewise, genetic lesions in pathway components we identified may lead to a block in cell death. Double mutants in apoptotic and LCD genes would allow testing of the combined contributions of these processes.

### The proteasome and LCD

As is the case with caspase proteases that mediate apoptosis (*Pop and Salvesen, 2009*), how the UPS induces LCD is not clear, and remains an exciting area of future work. That loss of BTBD-2, a specific E3 ligase component, causes extensive linker cell survival suggests that a limited set of targets may be required for LCD. Previous work demonstrated that BTBD2, the vertebrate homolog of BTBD-2, interacts with topoisomerase I (*Khurana et al., 2010*; *Xu et al., 2002*), raising the possibility that this enzyme may be a relevant target, although other targets may exist.

The UPS has been implicated in a number of cell death processes in which it appears to play a general role in cell dismantling, most notably, perhaps, in intersegmental muscle remodeling during metamorphosis in moths (*Haas et al., 1995*). However, other studies suggest that the UPS can have specific regulatory functions, as with caspase inhibition by IAP E3 ligases (*Ditzel et al., 2008*).

During *Drosophila* sperm development, caspase activity is induced by the UPS to promote sperm individualization, a process that resembles cytoplasm-specific activation of apoptosis (*Arama et al., 2007*). While *C. elegans* caspases are dispensible for LCD, it remains possible that they participate in linker cell dismantling or serve as a backup in case the LCD program fails.

Finally, the proteasome contains catalytic domains with target cleavage specificity reminiscent of caspases; however, inactivation of the caspase-like sites does not, alone, result in overt cellular defects (*Britton et al., 2009*), suggesting that this activity may be needed to degrade only specific substrates. Although the proteasome generally promotes proteolysis to short peptides, site-specific cleavage of proteins by the proteasome has been described (*Chen et al., 1999*). It is intriguing to speculate, therefore, that caspases and the proteasome may have common, and specific, targets in apoptosis and LCD.

## A pro-death developmental function for HSF-1

Our discovery that *C. elegans* heat-shock factor, HSF-1, promotes cell death is surprising. Heat-shock factors are thought to be protective proteins, orchestrating the response to protein misfolding induced by a variety of stressors, including elevated temperature. Although a role for HSF1 has been proposed in promoting apoptosis of mouse spermatocytes following elevated temperatures (*Nakai et al., 2000*), it is not clear whether this function is physiological. In this context, HSF1 induces expression of the gene Tdag51 (*Hayashida et al., 2006*). Both pro- and anti-apoptotic activities have been attributed to Tdag51 (*Toyoshima et al., 2004*), and which is activated in sperm is not clear. Recently, pathological roles for HSF1 in cancer have been detailed (e.g. *Mendillo et al., 2012*), but in these capacities HSF1 still supports cell survival.

Developmental functions for HSF1 have been suggested in which HSF1 appears to act through transcriptional targets different from those of the heat-shock response (*Jedlicka et al., 1997*), although target identity remains obscure. Here, we have shown that HSF-1 has at least partially non-overlapping sets of stress-induced and developmental targets. Indeed, typical stress targets of HSF-1, such as the small heat-shock gene *hsp-16.49* as well as genes encoding larger chaperones, like *hsp-1*, are not expressed during LCD, whereas *let-70*, a direct transcriptional target for LCD, is not induced by heat shock. Interestingly, the yeast *let-70* homologs *ubc4* and *ubc5* are induced by heat shock (*Seufert and Jentsch, 1990*), supporting a conserved connection between HSF and UBE2D2-family proteins. However, the distinction between developmental and stress functions is clearly absent in this single-celled organism, raising the possibility that this separation of function may be a metazoan innovation.

What distinguishes the stress-related and developmental forms of HSF-1? One possibility is that whereas the stress response appears to be mediated by HSF-1 trimerization, HSF-1 monomers or dimers might promote LCD roles. Although this model would nicely account for the differential activities in stress responses and LCD of the HSF-1(R145A) transgenic protein, which would be predicted to favor inactivation of a larger proportion of higher order HSF-1 complexes, the identification of conserved tripartite HSEs in the *let-70* and *rpn-3* regulatory regions argues against this possibility. Alternatively, selective post-translational modification of HSF-1 could account for these differences. In mammals, HSF1 undergoes a variety of modifications including phosphorylation, acetylation, ubiquitination, and sumoylation (*Xu et al., 2012*), which, depending on the site and modification, stimulate or repress HSF1 activity. In this context, it is of note that p38/MAPK-mediated phosphorylation of HSF1 represses its stress-related activity (*Chu et al., 1996*), and the LCD regulator SEK-1 encodes a MAPKK. However, no single MAPK has been identified that promotes LCD (E.S.B., M.J.K. unpublished results), suggesting that other mechanisms may be at play.

Our finding that POP-1/TCF does not play a significant role in LCD raises the possibility that Wnt signaling exerts direct control over HSF-1 through interactions with β-catenin. However, we have not been able to demonstrate physical interactions between these proteins to date (M.J.K, unpublished results).

Finally, a recent paper (*Labbadia and Morimoto, 2015*) demonstrated that in young adult *C. elegans*, around the time of LCD, global binding of HSF-1 to its stress-induced targets is reduced through changes in chromatin modification. Remarkably, we showed that chromatin regulators play a key role in *let-70* induction and LCD (J.A.M., M.J.K and S.S., manuscript in preparation), suggesting, perhaps, that differences in HSF-1 access to different loci may play a role in distinguishing its two functions.

## LCD and neurodegeneration

Previous studies from our lab raised the possibility that LCD may be related to degenerative processes that promote vertebrate neuronal death. Nuclear crenellation is evident in dying linker cells and in degenerating cells in polyQ disease (*Abraham et al., 2007*) and the TIR-1/Sarm adapter protein promotes LCD in *C. elegans* as well as degeneration of distal axonal segments following axotomy in *Drosophila* and vertebrates (*Osterloh et al., 2012*). The studies we present here, implicating the UPS and heat-shock factor in LCD, also support a connection with neurodegeneration. Indeed, protein aggregates found in cells of patients with polyQ diseases are heavily ubiquitylated (*Kalchman et al., 1996*). Chaperones also colocalize with protein aggregates in brain slices from SCA patients, and HSF1 has been shown to alleviate polyQ aggregation and cellular demise in both polyQ-overexpressing flies and in neuronal precursor cells (*Neef et al., 2010*). While the failure of proteostatic mechanisms in neurodegenerative diseases is generally thought to be a secondary event in their pathogenesis, it is possible that this failure reflects the involvement of a LCD-like process, in which attempts to engage protective measures instead result in activation of a specific cell death program.

# Materials and methods

## Strains

C. *elegans* strains were cultured using standard methods (*Brenner, 1974*) and were grown at 20°C unless otherwise indicated. Wild-type animals were the Bristol N2 subspecies. Most strains harbor one of two mutations that generate a high percentage of male progeny, *him-8(e1489)* IV or *him-5 (e1490)* V, as well as one of two integrated linker cell markers, *qIs56[lag-2*p::GFP]V or *nsIs65[mig-24*p::Venus] X. Other alleles and transgenes used are as follows:

LGI: *hsf-1(sy441)*, *lin-44(n1792)*, *mig-1(e1787)*, *lin-17(n3091)*, *unc-101(sy216)*, *gsk-3(nr2047)*, *pop-1 (q624)*, *pop-1(q645)*, *pop-1(hu9)*, *daf-16(mu86)*, *unc-13(e1091)*.

LGII: *mig-14(ga62)*, *lin-29(n333)*, *lin-29(n546)*, *mig-5(rh147)*, *cam-1(gm122)*, *cwn-1(ok546)*, *rrf-3 (pk1426)*, *drSi13[hsf-1*p::*hsf-1*-gfp], *drSi28[hsf-1*p::*hsf-1*(R145A)-GFP], *nsSi2[hsf-1*p::*hsf-1*(R145A)-GFP], *nsSi3[hsf-1*p::*hsf-1*(R145A)-GFP].

LGIII: *pqn-41(ns294)*, *wrm-1(ne1982)*, *lit-1(t512)*, *unc-32(e189)*, *mom-4(ne1539)*, *mom-4(or39)*, *unc-119(ed4)*.

LGIV: *siah-1(tm1968)*, *egl-20(n585)*, *cwn-2(ky756)*, *cwn-2(ok895)*, *let-70(ns770)*, *uba-1(it129)*, *btbd-2(gk474281)*.

LGV: *rde-1(ne219)*, *cfz-2(ok1201)*, *mom-2(ne834)*, *daf-21(p673)*.

LGX: *bar-1(ga80)*, *sek-1(ag1)*, *lin-18(e620)*.

## Transgenic strains

See *Supplemental file 2A*.

## MosSCI

Two additional lines of *hsf-1*p::*hsf-1* (R145A)-GFP were generated from pOG124 (a gift of T. Lamitina), by the 'direct' method, as previously described (*Frøkjaer-Jensen et al., 2014*). One line failed to exhibit *bar-1* mutant suppression, but also did not enhance *hsf-1(sy441)* survival, suggesting it was inactive, and was therefore not used in analysis. Inserts were verified by PCR and expression of HSF-1::GFP.

## Generation of *let-70(ns770)*, encoding LET-70(P61S)

*let-70(ns770)* was generated using co-CRIPSR-based CRISPR/Cas9-mediated genome editing as previously described (*Arribere et al., 2014*). pJA42 (Addgene, Cambridge, MA) was edited using PCR mutagenesis with a 'universal' forward primer (5'- GTTTTAGAGCTAGAAATAGCAAGTTAAAA TAAGGCTAG -3') and a *let-70* specific reverse primer (TTTCTAGCTCTAAAACATGGATAGTCTG TTGGGAAG CAAGACATCTCGCAATAG) to generate the *let-70* targeting vector. Single-stranded oligodeoxynucleotide 'repair' templates were ordered from Sigma for *let-70(ns770)* (5' TTAAATTTA TTTTTTTCCAATTTCGATCAATACCTTTGGTGGTTTAAATGAATAGTCTGTTGGGAAGTGGATAG TGAGGAAGAAGACACCTCCCTGATAGG 3') and *dpy-10(cn64)* (*Arribere et al., 2014*). N2 animals

were injected with the following mix: 50 ng pDD162 (Addgene), 25 ng pJA58 (*dpy-10* sgRNA, Addgene), 25 ng *let-70* targeting vector, 20 ng *dpy-10*(*cn64*) repair oligo, 20 ng *let-70*(*ns770*) repair oligo in 1x injection buffer (20mM potassium phosphate, 3mM potassium citrate, 2% PEG, pH 7.5).

F1 generation was screened for animals with a roller or dumpy-roller phenotype, indicating successful repair of the *dpy-10* break using the provided *dpy-10* oligonucleotide template, which were picked to individual plates. These animals were allowed to lay eggs and then genotyped for successful co-conversion of the *let-70* locus by PCR and Sanger sequencing. Non-roller F2 animals were then picked from successfully *let-70*-converted F1s and homozygosed for *let-70*(*ns770*) before outcrossing twice.

## Plasmid construction
See *Supplemental file 2B*.

## RNAi experiments
RNAi was performed by feeding on the strains indicated (*Blum et al., 2012*). Bleached embryos from gravid hermaphrodites were synchronized at the L1 stage by leaving them overnight in M9. L1s animals (30–50% of which were male) were added to each RNAi plate and grown for approximately 48 hr at 20–22°C. 0–2 hr adults were scored using a fluorescent dissecting scope (Leica). Clones were either newly created by cloning into the L4440 vector, or were already published clones from the Ahringer feeding library.

## RNAi-Resistant *let-70* cDNA
Total RNA was extracted using TRIzol (Theromfisher, Waltham, MA) using standard protocols. cDNAs were amplified from day one adult *Caenorhabditis briggsae* using the SuperScript II Reverse Transcriptase (Thermofisher). *C. briggsae let-70* cDNA with silent mutations was generated using GeneArt Gene Synthesis (Thermofisher) and cloned into plasmid using standard conditions. C85S point mutation was generated using Pfu turbo polymerase (Agilent, Santa Clara, CA) and DpnI digest (NEB, Ipswich, MA) using standard Quikchange protocol (Agilent).

## Germline transformation and rescue experiments
Germline transformation was carried out as previously described (*Mello et al., 1991*). For GFP expression analysis, all plasmids were injected into *unc-119*(*ed3*) III; *him-8*(*e1489*) IV hermaphrodites with *unc-119*(+) (*Maduro and Pilgrim, 1995*) as a transformation marker. All plasmids were injected at between 1–50 ng/ul. pBluescript (Agilent) was used to adjust the DNA concentration of injection mixtures if necessary. For rescue studies, animals were picked under a fluorescent dissecting microscope (Leica) the previous night as L3s with YFP- or mCherry-expressing linker cells to a new RNAi plate and scored the following day. Throughout, only correctly-migrated linker cells were used in determining survival percentages.

## Linker cell survival, migration, and GFP expression assays
Linker cell death was scored as previously described (*Blum et al., 2012*). Briefly, worms were synchronized by treating gravid hermaphrodites with alkaline bleach and allowing the eggs to hatch in M9 medium overnight. Synchronized L1s were released onto fresh NGM plates seeded with OP50 or HT115 *E. coli* containing the RNAi clone of interest, and maintained at 20°C. Animals were picked to a new plate as late L4s with a fully retracted tail tip with rays visible under the unshed L4 cuticle. Two hours later, newly molted adults were mounted on slides on 2% agarose-water pads, anaesthetized in 30 mM sodium azide or 5 mM tetramisole, and examined on a Zeiss Axioplan 2 or AxioScope A1 under Nomarski optics and widefield fluorescence at 40x or 63x. Images were acquired through a Zeiss AxioCam and the Axiovision software. The linker cell was identified by green fluorescence (from reporter transgenes) as well as by its location and morphology. A linker cell was scored as surviving if its nucleus was circular with an intact nucleolus, if the cell shape was not rounded, and if the cell had not shed any large blebs. All other cells were scored as dead or dying. Rescuing extrachromosomal arrays contained a *lag-2*p::mCherry construct, and, in an effort to prevent selection bias towards survival, males with mCherry-expressing linker cells were picked as L3s for scoring the following day as young adults, as above. Some Wnt pathway mutants exhibited two linker cells. For

these strains, animals with only one visible linker cell were picked as L3s to score the following day. Throughout, only correctly migrated cells that had reached the cloaca were used in determining survival percentages.

For GFP expression assays, 0–2 hr adults containing the *let-70*p::*let-70*::GFP (*nsIs241*) or *ubq-1*p::*let-70p*::GFP (*nsIs386*) transgenes were scored for the presence or absence of GFP expression in the linker cell. The fraction of animals expressing GFP was divided by the fraction of animals with surviving linker cells in order to obtain an accurate measure of linker cell expression. This method was verified by looking at GFP expression of reporters with a *lag-2*p::mCherry coinjection marker; results using the two different methodologies showed similar expression patterns.

## Electron microscopy

Just-molted (0–2 hr) *qIs56 him-5(e1490); bar-1(ga80)* or *let-70*(RNAi) adult males with surviving linker cells were imaged using a Zeiss Axioplan 2 compound microscope to measure the relative location of the linker cell within the worm using the AxioVision software (Zeiss). Animals were then fixed, stained, embedded in resin, and sectioned using standard methods (*Lundquist et al., 2001*). Images were acquired on an FEI TECNAI G2 Spirit BioTwin Transmission Electron Microscope with a Gatan 4K x 4K digital camera at The Rockefeller University EM Resource Center.

## Statistical methods

An unpaired t-test was used for GFP quantification in *rde-1* knockdown animals and in *let-70*p::*let-70*::GFP animals following heat shock. Fisher's Exact Tests were used for quantification of LCD experiments as well as quantification of GFP+ linker cells.

## Ubiquitination assay

*let-70* cDNA cloned into the vector pET28b(+) (Novagen, ) was transformed into BL21(DE3) cells using heat shock. Cells were induced overnight with 500 mM IPTG at 25°C. Purification was performed using a previously described protocol (*Sandu et al., 2010*). In vitro ubiquitination assay: A 40 µL reaction containing 3 µg each of purified *Drosophila* Uba1, Diap1, and ubiquitin (Gift from C. Sandu) were incubated with *C. elegans* His-LET-70 and reaction buffer (25 mM Tris, pH 7.5, 50 mM NaCl, 250 µM DTT, 4 mM ATP and 4 mM $MgCl_2$) for 30 min at 25°C (*Sandu et al., 2010*). One half of the reaction was run on an SDS-PAGE gel and stained with Coomassie Blue to visualize proteins.

## Heat-shock assays

Animals were cultured on 4 cm NGM agar plates seeded with *E. coli* OP50. These plates were sealed with parafilm, placed in a water bath at the indicated temperature for the indicated time, agar face down, and subsequently returned to the 20°C incubator, until animals were picked for scoring as above.

## Acknowledgements

We thank the members of the Shaham lab for discussions and comments on the manuscript, and Cori Bargmann, David Baillie, Dave Eisenmann, Mike Herman, Cynthia Kenyon, Stuart Kim, Rik Korswagen, Todd Lamitina, Lionel Pintard, and Aakanksha Singhvi, for reagents, and Sigi Benjamin, Cristi Sandu, and Hermann Steller for reagents and advice. Some strains were provided by the CGC, which is funded by NIH Office of Research Infrastructure Programs (P40 OD010440), and the National Bioresource Project of Japan. JAM was supported by the Rockefeller Women and Science Fellowship Program and NIH training grant CA09673. MJK was supported by NIH Medical Scientist Training Program grant T32GM07739. This work was supported by NIH grants NS081490 and HD078703 to SS.

## Additional information

### Funding

| Funder | Grant reference number | Author |
| --- | --- | --- |
| National Institutes of Health | HD078703 | Shai Shaham |
| National Institutes of Health | NS081490 | Shai Shaham |

The funders had no role in study design, data collection and interpretation, or the decision to submit the work for publication.

### Author contributions

MJK, JAM, Conception and design, Acquisition of data, Analysis and interpretation of data, Drafting or revising the article; MCA, ESB, Acquisition of data, Analysis and interpretation of data, Contributed unpublished essential data or reagents; MRS, Acquisition of data, Contributed unpublished essential data or reagents; YL, Acquisition of data, Analysis and interpretation of data; SS, Conception and design, Analysis and interpretation of data, Drafting or revising the article

### Author ORCIDs

Shai Shaham, http://orcid.org/0000-0002-3751-975X

## Additional files

### Supplementary files

• Supplementary file 1. Wnt pathway genes, UPS components, and SCF/BTB genes screened for linker cell death defects. (A) Wnt pathway genes and their effects on linker cell death. *All strains also contain the *qIs56* reporter transgene to visualize the linker cell, except for the *mom-2(ne834)* and *daf-21(p673)* strains which contained the *nsIs64* transgene. Allele numbers are in parentheses. † ± SEM. ‡Number of animals scored. ¶This strain also contained the *unc-13(e1091)* allele. §This strain had a severe migration defect which precluded accurate scoring of a survival phenotype. (B) RNAi against UPS components and effects on linker cell survival.*All animals contained *rrf-3(pk1426); him-8(e1489)* mutations and a *qIs56* reporter transgene to visualize the linker cell. †LC, linker cell. ± SEM. ‡Number of animals scored. (C) RNAi against SCF components/BTB domain proteins and linker cell survival. *All animals contained *rrf-3(pk1426); him-8(e1489)* mutations and a *qIs56* reporter transgene to visualize the linker cell. †LC, linker cell. ± SEM. ‡Number of animals scored.

• Supplementary file 2. Transgenic strains and plasmids used. (A) Transgenic strain allele number and relevant plasmids. (B) Plasmid names, descriptions, and construction.

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
