## [Decision Letter]

Thank you for submitting your work entitled "HSF-1 Activates the UPS to Promote
Non-Apoptotic Developmental Cell Death in *C. elegans*" for
consideration by *eLife*. Your article has been reviewed by two peer
reviewers, and the evaluation has been overseen by Oliver Hobert as a Reviewing Editor
and Naama Barkai as the Senior Editor.

The reviewers have discussed the reviews with one another and the Reviewing Editor has
drafted this decision to help you prepare a revised submission.

Summary:

The importance of the description of mechanistic aspects of linker cell death was
appreciated. However, there was a general sense that much more clarity and caution needs
to be applied to the description of experiments, results and, most importantly,
conclusions. These concerns can be addressed by some significant re-writing of a number
of the conclusions. No further experiments are required.

Essential revisions:

1) While the implication of a number of interesting genes in this pathway was
appreciated, there was a substantial concern about the interpretation of the genetic
interaction data. Throughout the manuscript, in a number of cases, genetic interaction
arguments were based on examining the enhancement of phenotypes based on the combination
of non-null alleles (or RNAi). Observed enhancements, even if synergistic in nature, do
not prove that genes act in the same pathway; they are equally consistent with genes
acting in separate pathways. The interpretation of all these genetic interaction tests
needs to be more correctly, and carefully, phrased.

2) The evidence that *hsf-1* controls *let-70* expression
is very preliminary and this should be made explicitly clear and conclusions be toned
down accordingly.

3) The authors try to describe a molecular-genetic framework governing the initiation
and execution of LCD. The initiation and execution process should be more clearly
defined and genes be more clearly categorized.

4) In the Introduction, the authors claimed that "our studies reveal intriguing
design similarities between LCD and apoptosis" This is overstated. How similar are
these two pathways? Depending on transcriptional factors and protein degradation? The
*hsf-1*/transcriptional evidence is weak and the involvement of
protein degradation relies on the implication of components of the proteasome pathway
but those can play roles in protein-degradation independent processes. While it is
interesting to speculate on difference, the conclusions need to be significantly toned
down.

---

## [Author Response]

*Essential revisions: 1) While the implication of a number of interesting genes
in this pathway was appreciated, there was a substantial concern about the
interpretation of the genetic interaction data. Throughout the manuscript, in a
number of cases, genetic interaction arguments were based on examining the
enhancement of phenotypes based on the combination of non-null alleles (or RNAi).
Observed enhancements, even if synergistic in nature, do not prove that genes act in
the same pathway; they are equally consistent with genes acting in separate pathways.
The interpretation of all these genetic interaction tests needs to be more correctly,
and carefully, phrased.*

We completely agree with this comment. Although none of our final conclusions depend on
the data of synergy between non-null alleles, we have, nonetheless changed the text
where such results may have appeared to be overreaching. The following specific changes
were made:

Results section:

“as would be predicted if HSF-1 functions” changed to “as might be predicted if HSF-1
functions”.

“indicating that these genes likely function together to promote LCD” changed to
“indicating that these genes likely function together, in sequence or in parallel, to
promote LCD”.

“We conclude that SIAH-1, CUL-3, and RBX-1 likely function together to promote LCD and
likely do so downstream of LET-70” changed to “We conclude that SIAH-1, CUL-3, and RBX-1
all function to promote LCD and likely do so downstream of LET-70”.

“consistent with the two genes acting in the same pathway” was removed.

*2) The evidence that hsf-1 controls let-70 expression is very preliminary and
this should be made explicitly clear and conclusions be toned down
accordingly.*

Our data demonstrate that (1) both
*hsf1* and *let-70* are required for linker cell death,
(2) that *let-70*::GFP
expression is induced upon cell death onset, (3)
that *hsf-1* is required for this transcriptional induction, (4) that a conserved HSF-1 binding site is present
upstream of *let-70* coding sequences, (5) that deletion of these upstream sequences inhibits induction of
*let-70*::GFP expression, and (6) that unlike all previously described LCD genes, *let70*
lesions cannot be rescued by an hsf-1 gain of function allele. We therefore feel that
the conclusion that *HSF-1* controls let-70 expression is reasonably well
substantiated.

Nonetheless, we agree with the reviewer comment in the sense that we have not
demonstrated that HSF-1 directly binds regulatory sequences upstream of the
*let-70* gene in vivo. However, this is a very tall order, as the
experiment would require performing chromatin IP from single linker cells, which is
currently a major technical challenge.

To avoid confusion about our claims, we have introduced the following changes:

Results section: “Taken together, our results strongly suggest that LET-70 functions
downstream of a linker-cell-specific non-canonical function of HSF-1 to promote LCD”
changed to “Taken together, our results suggest that LET-70 functions downstream of a
linker-cell-‐specific non-canonical function of HSF-1 to promote LCD. Our data also
suggest that other HSF-1 targets are likely relevant, and that let- 70 may be under the
control of additional regulators.”

*3) The authors try to describe a molecular-genetic framework governing the
initiation and execution of LCD. The initiation and execution process should be more
clearly defined and genes be more clearly categorized.*

We thank the reviewer for this comment, as we were indeed not consistent in our use of
these terms, and, in addition, the figure legend for our model had an error in color
assignment. We have now made several changes to the text, and have reserved use of the
terms “initiation” and “execution” only as they refer to the upstream or downstream
pathway components, respectively. The following specific changes have been made:

Figure 7 legend: Pathway color designations
changed to reflect the actual colors.

Introduction:

“Here we describe a molecular-genetic framework governing the initiation and execution
of LCD in *C. elegans*” changed to “Here we describe a molecular-genetic
framework governing LCD in *C. elegans*”.

“We demonstrate that LCD initiation is controlled” changed to “We demonstrate that LCD
is controlled”.

“increases just before LCD initiation” changed to “increases just before LCD onset”.

Results:

“dictate cell death initiation” changed to “dictate cell death onset”.

“in which organelle changes accompanying cell death initiation are evident” changed to
“in which organelle changes accompanying cell death are evident”.

“the parallel pathways controlling LCD initiation” changed to “the parallel pathways
controlling LCD onset”.

“*let-70* likely acts in the execution of LCD” changed to
“*let70* likely acts in promoting LCD”.

Discussion:

“the molecular identification of a LCD execution machinery” changed to “the molecular
identification of LCD components and their interactions”.

“may lead to a block in cell death execution” changed to “may lead to a block in cell
death”.

4) In the Introduction, the authors claimed that "our studies reveal
intriguing design similarities between LCD and apoptosis" This is overstated.
How similar are these two pathways? Depending on transcriptional factors and protein
degradation? The hsf-1/transcriptional evidence is weak and the involvement of
protein degradation relies on the implication of components of the proteasome pathway
but those can play roles in protein-degradation independent processes. While it is
interesting to speculate on difference, the conclusions need to be significantly
toned down.

We agree with the reviewer that we may have been too exuberant about the similarities in
our use of language. Nonetheless, the similarities we point out are apparent, and worth
noting, in our opinion. We have made the following changes in the text to tone down the
conclusions:

Introduction: “Our studies reveal intriguing design similarities” changed to “Our
studies reveal design similarities”.

Discussion: “The logic of the LCD pathway is strikingly similar” changed to “The logic
of the LCD pathway may be similar”.